# The Prediction of Batting Averages in Major League Baseball

**Sarah R. Bailey [1], Jason Loeppky [2] and Tim B. Swartz [1],***

[1]   Department of Statistics and Actuarial Science, Simon Fraser University, 8888 University Drive,
     Burnaby, BC V5A1S6, Canada; srbailey@sfu.ca
[2]   Department of Computer Science, Mathematics, Physics and Statistics, University of British Columbia
     Okanagan, 3187 University Way, Kelowna, BC VIV1V7, Canada; jason.loeppky@ubc.ca
*   Correspondence: tim@stat.sfu.ca

**Abstract:** The prediction of yearly batting averages in Major League Baseball is a notoriously difficult problem where standard errors using the well-known PECOTA (Player Empirical Comparison and Optimization Test Algorithm) system are roughly 20 points. This paper considers the use of ball-by-ball data provided by the Statcast system in an attempt to predict batting averages. The publicly available Statcast data and resultant predictions supplement proprietary PECOTA forecasts. With detailed Statcast data, we attempt to account for a luck component involving batting averages. It is anticipated that the luck component will not be repeated in future seasons. The two predictions (Statcast and PECOTA) are combined via simple linear regression to provide improved forecasts of batting average.

**Keywords:** big data; forecasting; logistic regression; PECOTA; Statcast

## 1. Introduction

Prediction in baseball is known to be notoriously difficult [1]. For example, consider the case of David Clyde who in 1973 was the number one draft pick in Major League Baseball (MLB) and was selected by the Texas Rangers. He was a much-celebrated prospect, and he first pitched in the major leagues at the tender age of 18 years. At 18, the famous sports magazine Sports Illustrated published an article [2] on Clyde, and he was described by some baseball scouts as the "best pitching prospect they had ever seen" [3]. Partly due to injuries, Clyde's career ended at the age of 26 with a dismal 18–33 winning record in MLB.

As an example of an established player whose future performance may not have been predicted accurately, consider the case of Albert Pujols. Pujols had 11 immensely successful seasons with the St. Louis Cardinals of MLB where he was rookie of the year in 2001, a three-time Most Valuable Player, a two-time Gold Glove winner and a nine-time All-Star. In St. Louis, Pujols had yearly batting averages that never dropped below 0.299 and was on a clear track to the Hall of Fame. In 2012, Pujols was traded to the Los Angeles Angels where he was given a 10-year $240 million contract (third largest in history at the time). In the past six seasons with Los Angeles (2012–2017), his batting averages have tumbled to 0.285, 0.258, 0.272, 0.244, 0.268, and 0.241. Some have claimed that Pujols is now MLB's worst player with four years remaining on his contract [4].

Although there are many performance statistics in baseball, our interest is the yearly prediction of batting averages. The batting average for a player is defined as the player's number of hits divided by their number of at-bats. There is considerable interest in batting averages. First, batting averages are important to fans. The sport of baseball is noted for its careful records, and batting averages have been recorded for all players for a very long time; the first statistics that are shown when looking at team rosters are typically batting averages. Second, batting averages are important to managers

regarding team composition and the negotiation of player salaries [5]. Thirdly, batting averages are also of interest to those who participate in fantasy sports [6]. In fantasy league baseball, participants "accumulate" players according to various sets of rules (e.g., restrictions on the number of players, restrictions on players of a given fielding position, budget restrictions where players are worth varying amounts, etc.). The participants obtain fantasy points that coincide with performance measures of their players such as hits, home runs, rbi's, wins, saves, etc. Participants are sometimes allowed to trade players, drop players and pick up additional players according to the particular fantasy rules. At the end of the contest, the participant with the most fantasy points is the winner, and often prizes are awarded. Fantasy sports have found their way into the "gambling" arena where online sites such as DraftKings (https://www.draftkings.com) and FanDuel (https://www.fanduel.com) have become extremely popular.

Perhaps the most well-known of the forecasting methods for batting average is the Player Empirical Comparison and Optimization Test Algorithm known as PECOTA [7]. Nate Silver developed this proprietary system in 2002–2003. Silver has gained recognition for many of his predictions that are of interest to the general public including the accurate predictions of US presidential elections [8]]. His methods are statistically based, and in the case of PECOTA, the approach relies on relevant baseball features such as age, position, body type, past performance, etc. However, due to the proprietary nature of PECOTA, the exact calculations of the predictions are unknown. Baseball Prospectus purchased the PECOTA system in 2003, and since that time, the annual editions of Baseball Prospectus (e.g., [9]) have been the sole source of the PECOTA predictions. We note that PECOTA also provides forecasts for other MLB baseball statistics in addition to batting average.

Our research question is whether the prediction of batting averages can benefit from the wave of data that is now available via the big data revolution in baseball. Specifically, we are interested in whether the detailed ball-by-ball data provided by the Statcast system [10] contains information that is relevant to the prediction of batting averages. Sportvision created the Statcast system, and it was introduced in the 2015 regular season for every MLB game. Statcast provides publicly available data on every pitch that is thrown and is based on both Doppler radar (for tracking balls) and video monitoring (for tracking players). Given that PECOTA is complex and proprietary, it would be interesting whether a simple and fully prescribed prediction system can be developed based on data that is freely available to everyone.

This paper is based on the work of [11] where the motivating idea is the detection of a luck component in a batter's season which may not persist and is not predicted to continue in subsequent seasons. The luck component may be either good luck or bad luck. In using the term "luck", we recognize that luck may be explained as a consequence of regression towards the mean [12]. Luck is established by looking at the characteristics of the detailed player at-bats using the Statcast data. The Statcast variables that we consider in estimating the probability of a hit are the launch angle, the exit velocity and the distance that the ball is hit. In addition, we supplement that Statcast dataset with other publicly available datasets that provide information on the handedness of hitters and their footspeed.

In the literature, various defensive independent pitching statistics (DIPS) have been proposed. These statistics are also rooted in the recognition of the role of luck in baseball [13]. In an oversimplification of DIPS, the idea is that a pitcher has ultimate control over strikeouts, walks, home runs and hit by pitches whereas other batting outcomes are subject to fielding and luck. Therefore, DIPS statistics focus on the controllable outcomes. [14] has also considered luck as an element of batting and pitching performance by proposing random-effects models on components of performance where the components have different levels of luck. Albert's components are characterized by at-bats that are distinguished according to strike-outs, home runs and balls put into play. Shrinkage ideas are also pertinent to the luck phenomena, and various shrinkage methods have been proposed in the literature for prediction in baseball. For example, [15] develop a Bayesian model that is used to predict home run totals.

In Section 2, we carry out a data analysis that is based on two years (2015–2016) of Statcast data and auxiliary data. First, we check the integrity of the 2015 data and observe that apart from some missingness, the data is reliable. Based on the Statcast data for all players in 2015, logistic regression models are used to estimate the probability of a hit. From these probabilities, we obtain predicted Statcast batting averages for 2016. We then use the 2016 Statcast predictions, the 2016 PECOTA predictions and the actual 2016 batting averages to obtain combined (Statcaset and PECOTA) 2017 predictions. In Section 3, the 2017 predictions are assessed where we observed that the combined predictions are an improvement over straight PECOTA. We also derive approximate standard errors for the combined predictions. We conclude with a short discussion in Section 4.

The main contribution of our paper is the demonstration that using publicly available data (Statcast) and some simple tools, one can investigate notoriously difficult prediction problems in MLB. Moreover, the methods that are proposed here already compete with state-of-the-art proprietary methods. With more investigation, it would not be surprising that modifications could lead to improved predictions.

## 2. Data Analysis

Although the data analysis procedure is not complicated, it consists of various steps. In these steps, we have been careful not to error by making double use of the data (i.e., using the same data for both model fitting and model assessment). The data analysis steps are summarized in Figure 1.

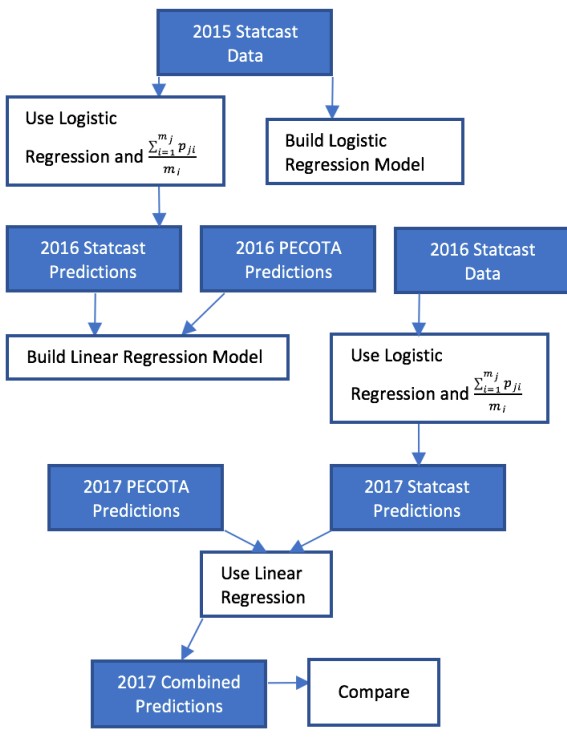

**Figure 1.** Overview of the analysis.

We remark that Statcast data is "big data". For every pitch that is thrown, Statcast records over 30 variables related to defensive, offensive and pitching outcomes. With 30 teams in MLB, 162 matches in a season and roughly 150 pitches per game, this leads to over 10 million data values collected per season.

### 2.1. Using 2015 Data and Predictions

As Statcast is a relatively new tracking system, we began by investigating aspects of its reliability. Each case (each thrown ball) in the Statcast dataset has many associated variables including the

identification of the batter, the identification of the pitcher and the properties of the thrown ball. We used the event variable in the Statcast dataset to determine the outcome of each at-bat (e.g., fielding error, strikeout, double play, etc.) In doing so, we used the resultant outcomes to calculate the batting average for batters during the 2015 regular season. We then compared the batting averages and the number of bats for five randomly selected players with the reported values from www.baseball-reference.com. The five players were Aramis Ramirez, Troy Tulowitzki, Trevor Plouffe, Omar Infante, and Bobby Wilson. In each case, both their number of at-bats and the number of hits matched exactly.

Our first decision in the data analysis was to restrict the prediction of batting averages to players for whom we had adequate data in the previous season. Without sufficient at-bats, there may be great variation in the predicted batting averages. Therefore, for the 2015 regular season, we only considered players who had at least 200 at-bats. We were left with 334 players who met the threshold. Consequently, we used 84.1% of all at-bats in the 2015 dataset (i.e., 140,896 of the 167,606 at-bats).

In the next step of our procedure, we want to estimate the probability of a hit based on the characteristics of the particular at-bat. The variables that we used in assessing the characteristics of an at-bat were the exit velocity ($x_1$) the launch angle ($x_2$) and the distance that the ball was hit ($x_3$). Our intuition is that these are physically meaningful variables regarding the probability of a hit. For example, we believe that the harder the ball is hit (i.e., the greater the exit velocity), the greater the chance of a hit. It also seems that the relationship between $p = \text{Prob}(\text{hit})$ and the three variables are not necessarily linear and that interactions may exist. We also consider two further variables that are provided by auxiliary datasets. We introduce the indicator variable $x_4$ which characterizes the handedness of the batter with the idea that a left-handed batter is closer to first base than a right handed batter, and may, therefore, be advantaged in running out a ground ball. We also introduce the continuous variable $x_5$ which is the footspeed of players. The variable $x_5$ is obtained from the webpage www.baseballsavant.mlb.com/sprint_speed_leaderboard where the idea again is that faster runners may be advantaged in running out a ground ball. These observations are therefore suggestive of a logistic regression model where logit($p$) is regressed against covariates originating from $x_1$, $x_2$, $x_3$, $x_4$ and $x_5$.

However, before proceeding with logistic regression, missing Statcast data is a concern since missingness can introduce bias. There are some batting outcomes for which covariates are systematically missing. For example, when there is a strikeout, $x_1$, $x_2$, and $x_3$ are always missing since the ball is not put into play. Therefore, in the case of strikeouts, we assigned $p = 0$. There were 29,713 strikeouts of the 140,896 at-bats in the restricted dataset (i.e., 21.1%).

High angled pop-outs are another case that requires special attention. Our investigation found that radar sometimes loses track of the ball, and in these instances, imputed values of $x_1$, $x_2$, and $x_3$ were provided, and these imputed values cannot be deemed reliable [16]. Although this is not the same sort of missingness as with strikeouts, almost all pop-outs result in outs. Therefore, in the case of pop-outs, we also assigned $p = 0$. There were 7341 pop-outs of the 140,896 at-bats in the restricted dataset (i.e., 5.2%).

There was one more remaining case of systematic missingness that proved problematic, and it involved ground balls. With ground balls, the hit distance variable $x_3$ was missing when the ball was stopped by an infielder before it could reach its true hit distance. Our solution to this problem was to take all ground balls and carry out a logistic regression of $p$ solely against the exit velocity variable $x_1$ given by Statcast and the auxiliary variables $x_4$ and $x_5$. There were 50,717 ground balls of the 140,896 at-bats in the restricted dataset (i.e., 36.0%). The fitted logistic regression model was

$$\log\left(\frac{p}{1-p}\right) = -4.515 + 0.039x_1. \tag{1}$$

To compare (1) versus our intuition, consider batted balls with exit velocities $x_1 = 20$ mph and $x_1 = 120$ mph. The fitted probabilities of a hit are 0.023 and 0.541, respectively. As is reasonable, the

ball which is hit harder has the greater probability of a hit. The AIC fit diagnostic for (1) is 57011 whereas it is 57170 for the null model with only the intercept term.

In the main logistic regression, we removed observations corresponding to strikeouts, pop-outs and ground balls. We were left with only 3280 missing values which we attribute to missing at random. The missing values constitute only 2.3% of the restricted dataset involving 140,896 at-bats. Therefore, the complete data logistic regression involves 49,845 of the 140,896 at-bats (i.e., 35.4%). We characterize these remaining observations as fly-balls. We considered all third-order covariates involving $x_1$, $x_2$, and $x_3$ since there could be complex interactions between some of the covariates. For example, with launch angle $x_2$, it is intuitive that balls hit at a higher angle relative to the ground will have a greater probability of resulting in a hit. However, there is a caveat in that if the ball is a little too high, then it will provide the fielder with more time to reach the ball and make a catch. Looking ahead to Equation (2), we indeed see that the covariate $x_2^3$ has a negative coefficient. We did not include $x_4$ (handedness)and $x_5$ (footspeed) as our physical understanding of baseball suggests that these variables should not be relevant in the prediction of hits from fly-balls. Then, after removing covariates with correlations exceeding 0.95, we carried out stepwise regression on the remaining variables. We obtained the fitted logistic regression model

$$\log\left(\frac{p}{1-p}\right) = -1.370 + 0.0215x_1 + 0.00312x_3 - 0.0000265x_2^3 - 0.000152x_2x_3, \tag{2}$$

where all included variables are highly significant. We note that each of the estimated parameters of the first order terms have signs that correspond to our intuition. The AIC fit diagnostic for (1) is 54587 whereas it is 68801 for the null model with only the intercept term.

Now, the final step involving the 2015 data involves the 2016 Statcast predictions. The simple philosophy is that players will behave similarly in 2016 as they did in 2015 with the exception that their "luck" is modified. Consider the $j$th player who had $m_j$ at-bats in 2015. In his $i$th at-bat during the 2015, we determine the probability of a hit $p_{ji}$ where $p_{ji} = 0$ if his at-bat was either a strikeout or a pop-out, $p_{ji}$ is calculated according to (1) if his at-bat was a ground ball, and $p_{ji}$ is calculated according to (2), otherwise. Therefore, the Statcast predicted batting average for player $j$ in the 2016 season is given by

$$y_j^{(S)} = \frac{\sum_{i=1}^{m_j} p_{ji}}{m_j}. \tag{3}$$

Figure 2 provides a scatterplot of the 2016 Statcast predictions versus the 2016 PECOTA predictions. We observe a similarity between the two sets of predictions as they appear scattered about the straight line $y = x$. We do note that the Statcast predictions are slightly more extreme as they have both larger and smaller predictions than PECOTA.

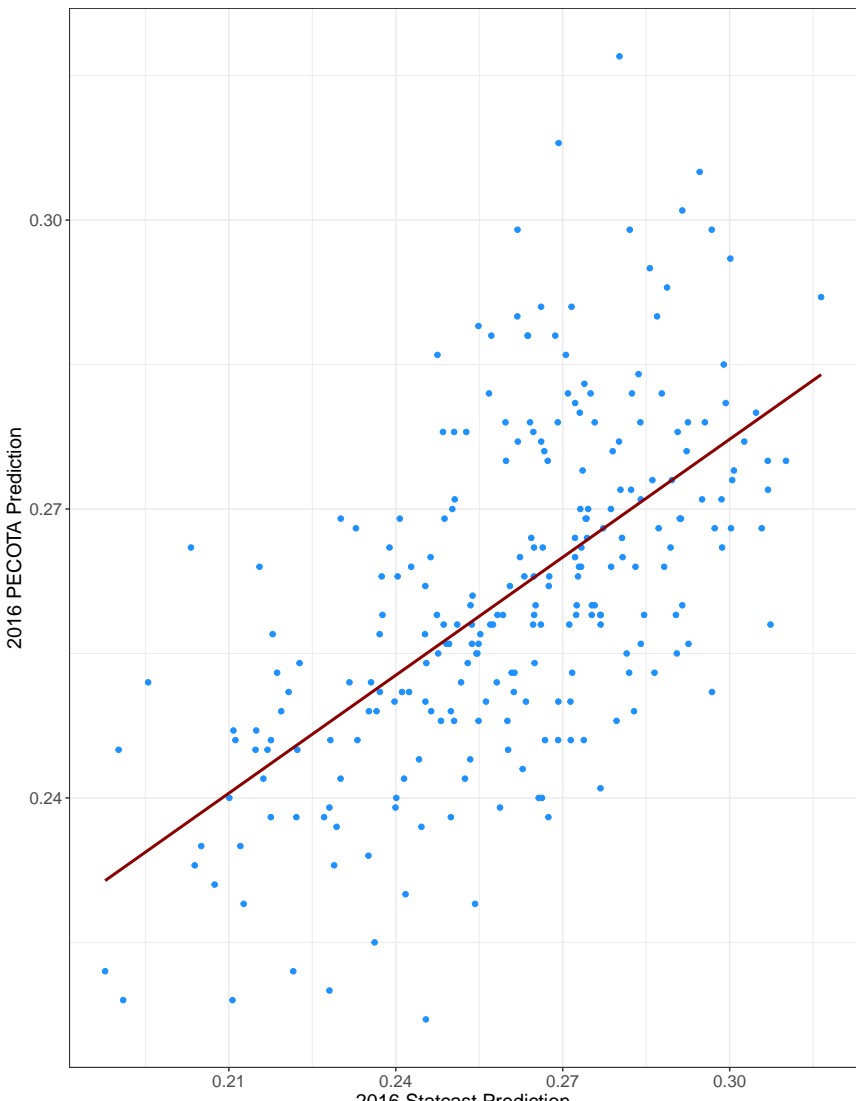

**Figure 2.** Statcast and PECOTA predictions for 2016 season. The line $y = x$ has been added to aid in interpretation.

*2.2. Using 2016 Data and Predictions*

We recall that the Statcast 2016 predictions are not sophisticated in that they do not consider important variables such as age, injuries and player speed. However, what the Statcast 2016 forecasts do implicitly include is information concerning the luck of the player during the 2015 season. If he was lucky in 2015 (had more hits than implied by (3)), then his 2016 prediction will be less than his actual 2015 batting average. On the other hand, if he was unlucky in 2015 (had fewer hits than implied by (3)), then his 2016 prediction will be greater than his actual 2015 batting average.

There is a wisdom of the crowd philosophy (also related to model averaging) that suggests that information from various sources is often superior to information from a single source. Following these beliefs, we consider the simple linear regression model

$$y_j^{(A)} = \beta_0 + \beta_S y_j^{(S)} + \beta_P y_j^{(P)} + \epsilon_j, \tag{4}$$

where $y_j^{(A)}$ is the actual batting average in 2016 for player $j$, $y_j^{(S)}$ is the 2016 Statcast prediction for player $j$, $y_j^{(P)}$ is the 2016 PECOTA prediction for player $j$ and $\epsilon_j$ is a random error term.

The least squares fit of the simple linear regression model (4) leads to

$$y_j^{(C)} = 0.0257 + 0.2577 y_j^{(S)} + 0.6482 y_j^{(P)}, \tag{5}$$

where we refer to $y_j^{(C)}$ as the combined predictor for player $j$ in 2016. In (5), we observe the fit diagnostic $R^2 = 0.29$. Of course, it would not be fair to assess $y_j^{(C)}$ versus $y_j^{(A)}$ since that would be a violation of data analytic principles where the same data are used for both model fit and model assessment. From (5), we observe that the existence of the intercept term implies that there is some bias (although small) in the Statcast and PECOTA predictors. We also observe that more weight is placed on the PECOTA predictor than on the Statcast predictor.

## 3. Assessing 2017 Predictions

To assess the combined predictor (5), we shift our comparison to the subsequent year, 2017. Accordingly, we determine the Statcast predictors $y_j^{(S)}$ for 2017 where we again evaluate (3). However, this time the calculation of (3) was based on the $p_{ji}$ from the 2016 regular season. The $p_{ji}$ were obtained from the same fitted logistic regression Equations (1) and (2) but we used the 2016 covariates $x_1$, $x_2$ and $x_3$ for each at-bat.

Once we obtained the Statcast predictors $y_j^{(S)}$ for 2017, we plugged those values into (5) together with the 2017 PECOTA predictors $y_j^{(P)}$. The fitted regression (5) gives the 2017 combined predictors $y_j^{(C)}$ where we emphasize that $y_j^{(C)}$ has not used 2017 data in any way.

Given $y_j^{(S)}$, $y_j^{(P)}$, $y_j^{(C)}$ and $y_j^{(A)}$ for 2017, there are various comparisons of interest. The first diagnostic we consider is mean absolute error

$$\text{MAE} = \frac{1}{n} \sum_{j=1}^{n} |y_j^{(A)} - y_j^{(i)}|, \tag{6}$$

where $n = 333$ is the number of players that are considered in 2017 and $i = S, P, C$. MAE measures the average discrepancy between actual batting average and the particular prediction method. In Table 1, we provide the MAE values for 2017 and other prediction statistics. Referring back to the comments in the Introduction, we observe that the prediction of batting averages is a difficult task. For example, even with the well-established PECOTA system, the average error in predicting a batting average is roughly 21 batting points. This level of error is a meaningful difference in the perceived quality of a batter. The next thing that we observe from Table 1 is that there is not a great difference between the three predictions methods; the combined predictor is best, PECOTA slightly trails, and then Statcast is the weakest.

**Table 1.** Comparison of the three prediction methods in 2017. We include mean absolute error (MAE), 95% CI for MAE based on 100 bootstrap samples, ME (mean error), average prediction, standard deviation of the prediction and prediction percentiles.

| Method | MAE | MAE CI | ME | Avg | Sd | 5th Perct | 95th Perct |
|--------|------|-----------------|---------|-------|-------|-----------|-----------|
| Statcast | 0.0236 | (0.0229,0.0233) | −0.0009 | 0.260 | 0.026 | 0.212 | 0.299 |
| PECOTA | 0.0209 | (0.0205,0.0209) | −0.0017 | 0.261 | 0.018 | 0.234 | 0.291 |
| Combined | 0.0208 | (0.0198,0.0202) | 0.0000 | 0.262 | 0.017 | 0.233 | 0.287 |

To get a visual sense of the 2017 Statcast and PECOTA predictions, Figure 3 provides scatterplots of the predictions versus the actual batting averages. We observe a little more variation in the Statcast predictions. However, the overall impression is that the two prediction approaches are comparable; this reinforces the message that predictions can be made using simple methods and publicly available data (Statcast) that compete with sophisticated and proprietary methods (PECOTA).

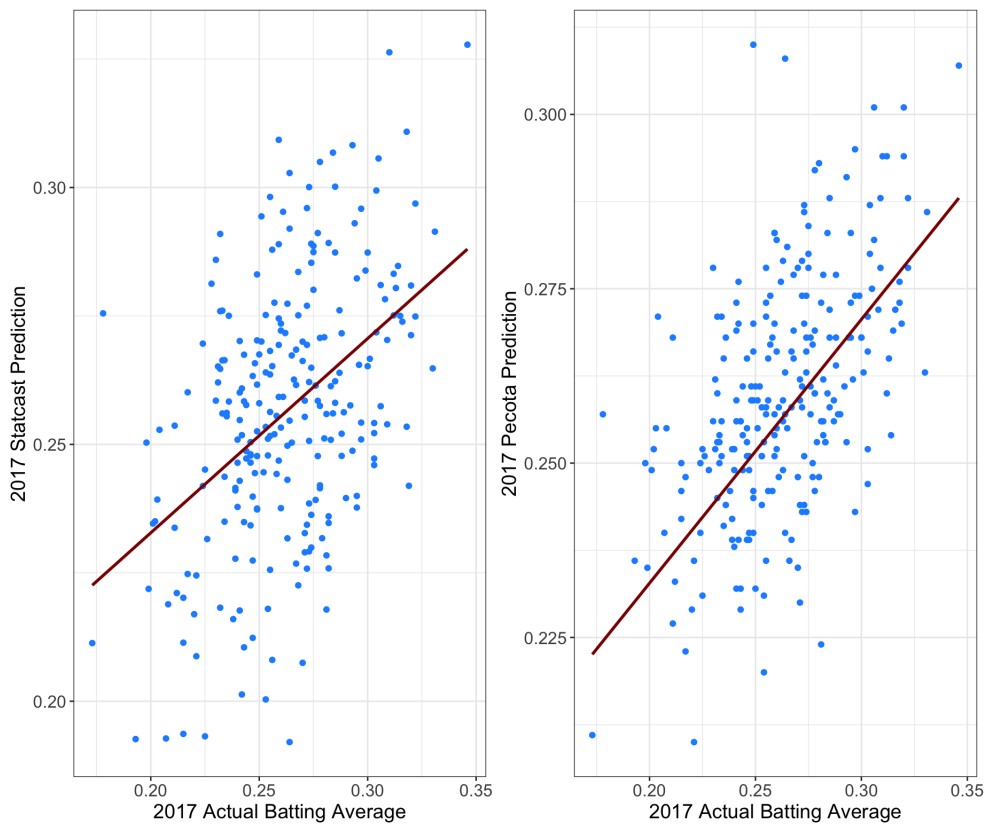

**Figure 3.** Scatterplots of the Statcast and PECOTA predictions for the 2017 season plotted against actual batting averages. Simple linear regression lines have been added to both plots.

However, our initial concern was not whether Statcast predictions would be better than PECOTA. We anticipated that Statcast would be weaker because it did not take into account important variables. The real question was whether Statcast could assist PECOTA through the construction of the combined predictor $y_j^{(C)}$. Recall that Statcast data is freely and publicly available, and therefore can be used by anyone.

We now look at a second diagnostic for comparison purposes. We calculated the percentage of the time that the combined predictors $y_j^{(C)}$ are closer to the actual batting average $y_j^{(A)}$ than the PECOTA values $y_j^{(P)}$. It turns out that the combined predictor is closer 56% of the time, a meaningful difference. Together, these two diagnostics suggest that when Statcast is bad, it can be quite bad. This observation may have been anticipated by the more extreme predictions observed in Figure 2.

Finally, we investigate the two players whose 2017 predictions were the poorest. Tyler Saladino of the Chicago White Sox had comparable predictions of $y^{(C)} = 0.260$, $y^{(P)} = 0.257$ and $y^{(S)} = 0.264$, yet his actual batting average was $y^{(A)} = 0.178$. Saladino was plagued with injuries during 2017 which contributed to a lower than expected performance. He had only 253 bats which also contributed to the variability of his performance.

At the other end of the scale, Avisail Garcia, also of the Chicago White Sox had predictions of $y^{(C)} = 0.262$, $y^{(P)} = 0.263$ and $y^{(S)} = 0.256$, yet is actual batting average was an outstanding $y^{(A)} = 0.330$. Garcia had a breakout 2017 season which seemed unexpected. He batted only 0.257 and 0.245 in the 2015 and 2016 regular seasons.

*Approximate Standard Errors*

Predictions are rarely exact and it is, therefore, useful to have a sense of prediction error. Since our prediction approach involves multiple steps, it is difficult (impossible) to obtain an exact standard

error for the combined predictor (5). Instead, we consider an approximation of the standard error where we assume that the coefficients in (5) are known. This leads to the variance expression

$$\mathrm{Var}(y_j^{(C)}) \approx (0.2577)^2 \mathrm{Var}(y_j^{(S)}) + (0.6428)^2 \mathrm{Var}(y_j^{(P)}) + 2(0.2577)(0.6428)\mathrm{Cov}(y_j^{(S)}, y_j^{(P)}), \qquad (7)$$

In (7), we estimate $\mathrm{Var}(y_j^{(P)}) = 0.00029$ by squaring the sample standard deviation of the 2016 PECOTA predictions. We estimate $\mathrm{Cov}(y_j^{(S)}, y_j^{(P)}) = 0.00030$ by calculating the sample covariance from the values displayed in Figure 2. Referring to (3), the Statcast variance is estimated by $\mathrm{Var}(y_j^{(S)}) = (1/m_j)^2 \sum_{i=1}^{m_j}(\mathrm{SE}(p_{ji}))^2$ where $\mathrm{SE}(p_{ji})$ is the approximate standard error of $p_{ji}$ obtained from logistic regression output. Having substituted the aforementioned estimates in (7), we then obtain the approximate standard error $\mathrm{SE}(y_j^{(C)}) = \sqrt{\mathrm{Var}(y_j^{(C)})}$.

In Table 2, we provide the approximate standard errors of the combined predictor (5) for the first 10 batters (alphabetical) during the 2017 season. We observe that the standard errors are roughly 16 batting points; this is slightly favorable to the PECOTA sample standard deviation which is roughly 17 batting points.

**Table 2.** Predictions and approximate standard errors for the first 10 batters (alphabetically) in the 2017 season.

| Batter | $y_j^{(C)}$ | $\mathrm{SE}(y_j^{(C)})$ |
|---|---|---|
| Aaron Hicks (New York Yankees) | 0.245 | 0.0162 |
| Adam Duval (Cincinnati Reds) | 0.240 | 0.0157 |
| Adam Jones (Baltimore Orioles) | 0.267 | 0.0160 |
| Adam Lind (Seattle Mariners) | 0.269 | 0.0164 |
| Adam Rosales (San Diego Padres) | 0.224 | 0.0157 |
| Addison Russell (Chicago Cubs) | 0.241 | 0.0159 |
| Adeiny Hechavarria (Miami Marlins) | 0.262 | 0.0158 |
| Adrian Gonzalez (Los Angeles Dodgers) | 0.272 | 0.0158 |
| Adrian Beltre (Texas Rangers) | 0.292 | 0.0158 |
| Albert Pujols (Los Angeles Angels) | 0.269 | 0.0158 |

## 4. Discussion

We have observed a big data phenomenon; that the detailed information provided by the Statcast dataset in MLB can help improve the well-established PECOTA forecasts. The assistance is facilitated through the combined predictor $y_j^{(C)}$ in (5).

It is worth asking how prediction might be improved. As previously noted, the Statcast system is relatively new with only three seasons of data, 2015–2017. As more data comes on board, it may be possible to improve estimation. This may be possible by improving the logistic regression equations given by (1) and (2). It may also be possible to include auxiliary information to Statcast data (e.g., age, injuries) to improve the Statcast prediction so that it is comparable to the proprietary predictions given by PECOTA. It may also be possible to improve predictions using various machine learning methods such as random forests and neural networks instead of the basic regression techniques used in this paper.

In MLB, we chose Jose Altuve of the Houston Astros as a player of interest as he had the highest batting average (0.346) in 2017. Our proposed Statcast methods predicted a 2018 batting average of 0.269 for Altuve. Our 2018 prediction for Altuve suggests that he benefitted from luck in 2017. Altuve's actual batting average in 2018 was 0.316; therefore although our prediction was on the low side, Altuve did not repeat his remarkable performance from 2017. In this example, PECOTA did extremely well by predicting a 0.315 batting average for Altuve in 2018.

**Author Contributions:** S.R.B., J.L., T.B.S. contributed to the research efforts of the paper with all of the computation undertaken by S.R.B. All authors have read and agreed to the published version of the manuscript.

**Funding:** This research received no external funding.

**Acknowledgments:** Swartz and Loeppky received partial support from the Natural Sciences and Engineering Research Council of Canada (NSERC). The authors thank three anonymous reviewers whose contributions have improved the paper.

**Conflicts of Interest:** The authors declare no conflicts of interest.

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
