# Peer review of "The Prediction of Batting Averages in Major League Baseball"

_stats, doi:10.3390/stats3020008_

Round 1
Reviewer 1 Report
In this paper a method for predicting batting average in Major Leage Baseball based on Statscast data is presented and discussed. Results are mixed with PECOTA method to improve predicton.
I would kindly ask to the authors these questions:
- Why logistic model was chosen instead of other machine learnine method such as logistic regression?
- Which is R2 of each regression, in particular equations (1), (2) and (5)
- How variables of Eq. (2) were chosen, in particular x23 and x2x3?
- Lines 192-195. On my opinion science paper should not talk about "lucky" or "unlucky". The phenomena described in these lines is calles regression to the average (for example here https://academic.oup.com/ije/article/34/1/215/638499 or simply https://en.wikipedia.org/wiki/Regression_toward_the_mean), when a player will have a successfull season the next one "probably" it will be unsuccessfull thanks to regression to the mean. Could you model it?
Author Response
Reviewer #1
Thank-you for your comments. They are helpful and we have
added an acknowledgment to all three reviewers.
1. Yes, I think it is generally accepted that some machine learning
techniques such as random forests and neural networks lead to improved
predictions over logistic regression. However, regression techniques
generally have an advantage in terms of interpretation. For example,
if a covariate x increases by 1 unit, then we can expect the left hand
side of the regression equation to increase by \beta (the parameter
coefficient of x). In our application, we wanted to make sure that the
prediction made physical sense - e.g. it should be the case that a ball
which is hit harder has a greater chance of being a hit. But, we take
your time and have added a comment to this effect in the Discussion.
2. As requested, we have included the R^2 diagnostic corresponding to the
model (5). Logistic regression does not have a natural R^2
diagnostic, and instead, we have provided AIC values for (1) and (2).
3. The reason why we investigated third order terms (eg x_2^3) relates
back to comment (1). An advantage of machine learning techniques is that
they do not rely on linear relationships. And although we have domain
knowledge of baseball, we realized that there could be complex interactions
between some of the covariates. For example, with launch angle x_2, it is
intuitive that balls hit at a higher angle relative to the ground will
have a greater probability of resulting in a hit. However, there is a
caveat in that if the ball is a little too high, then it will provide
the fielder with more time to reach the ball and make a catch. This is
why the cubic term x_2^3 has a negative coefficient. We have added several
sentences in the paragraph preceding equation (2) to expand on these issues.
4. Yes, regression to the mean is a good way of explaining ``luck''.
Thank-you. Luck is a repeated theme in the paper; we have therefore
added some explanation and the suggested reference in the first instance
of luck in the Introduction.
Reviewer 2 Report
This paper presents a the prediction framework for batting averages with PECOTA and Statcast. I think the paper is well written, while I still have several concerns.
1.It seems with more detailed the data, the prediction accuracy increase marginally, as shown in Table 1. Is there any confidence interval can be shown in Table 1?
- This paper reads like a project report, please summarize and highlight the contribution in the introduction sections.
- Figure 2, please add x=y line as reference
- The authors claim they are using big data, please introduce the data sets and quantify how big the dataset is.
Author Response
Reviewer #2
Thank-you for your comments. They are helpful and we have
added an acknowledgment to all three reviewers.
1. Yes, there is very little improvement using our methods. The main
message is that using publicly available data (Statcast) and some
simple tools, we can do about as well as state-of-the-art proprietary
methods. We have now provided some standard errors for the entries in
Table 1 using a bootstrap procedure.
2. In the last paragraph of the Introduction, we now provide an
overview of the main contribution of the paper.
3. Yes, thank-you. We have added the line y=x to Figure 2.
This aids in interpretation.
4. In the second paragraph of Section 2, we now describe the
scope of the Statcast data in terms of big data.
Reviewer 3 Report
see attached file

Author Response
Reviewer #3
Thank-you for your comments. They are helpful and we have
added an acknowledgment to all three reviewers.
1. Some new columns have been added to Table 1 as requested.
2. We have now included mean error (ME) in Table 1.
3. We have provided a new Figure 3 which provides plots
of forecasts versus actual for both PECOTA and Statcast
predictions. This allows the reader to assess the predictions.
4. The regression lines corresponding to the two scatterplots
in (3) have been added.
Minor Points
1. Yes, the term ``missingness'' is common. We have retained it.
2. We take your point. It seems unusual that if the response depends
on the interaction term x2*x3, then it ought to also depend on x2.
However, in this application, x2 is not signficant. We have googled
this question, and there seems to be conflicting advice on whether
to report lower order terms that are not significant. On the one
hand is your point. On the other hand, there is the preference for
model simplicity. Since we have some cubic terms in our model, we
have chosen to not introduce the lower order terms as this would
lead to a substantially larger model with some coefficients that
may not be intuitive.